

# Cost-benefit analysis for invasive species control: the case of greater Canada goose *Branta canadensis* in Flanders (northern Belgium)

Nikolaas Reyns[1,*], Jim Casaer[2], Lieven De Smet[2], Koen Devos[2], Frank Huysentruyt[2], Peter A. Robertson[3], Tom Verbeke[4] and Tim Adriaens[2,*]

[1] Faculty of Economics and Business Administration, University of Ghent, Ghent, Belgium
[2] Research Institute for Nature and Forest (INBO), Brussels, Belgium
[3] Centre for Wildlife Management, Newcastle University, Newcastle, United Kingdom
[4] Research Centre for Economics and Corporate Sustainability, University of Leuven, Brussels, Belgium
[*] These authors contributed equally to this work.

## ABSTRACT

**Background**. Sound decisions on control actions for established invasive alien species (IAS) require information on ecological as well as socio-economic impact of the species and of its management. Cost-benefit analysis provides part of this information, yet has received relatively little attention in the scientific literature on IAS.

**Methods**. We apply a bio-economic model in a cost-benefit analysis framework to greater Canada goose *Branta canadensis*, an IAS with documented social, economic and ecological impacts in Flanders (northern Belgium). We compared a business as usual (BAU) scenario which involved non-coordinated hunting and egg destruction with an enhanced scenario based on a continuation of these activities but supplemented with coordinated capture of moulting birds. To assess population growth under the BAU scenario we fitted a logistic growth model to the observed pre-moult capture population. Projected damage costs included water eutrophication and damage to cultivated grasslands and were calculated for all scenarios. Management costs of the moult captures were based on a representative average of the actual cost of planning and executing moult captures.

**Results**. Comparing the scenarios with different capture rates, different costs for eutrophication and various discount rates, showed avoided damage costs were in the range of 21.15 M€ to 45.82 M€ under the moult capture scenario. The lowest value for the avoided costs applied to the scenario where we lowered the capture rate by 10%. The highest value occurred in the scenario where we lowered the real discount rate from 4% to 2.5%.

**Discussion**. The reduction in damage costs always outweighed the additional management costs of moult captures. Therefore, additional coordinated moult captures could be applied to limit the negative economic impact of greater Canada goose at a regional scale. We further discuss the strengths and weaknesses of our approach and its potential application to other IAS.

Corresponding author
Tim Adriaens, tim.adriaens@inbo.be

## INTRODUCTION

Invasive alien species (IAS) can severely impact on society causing ecological, economic and human health impacts (e.g., *Olson, 2006*; *Pejchar & Mooney, 2009*; *Vila et al., 2010*; *Schindler et al., 2015*; *Roy et al., 2016*). Invasive species are sometimes intentionally introduced to exploit economic benefits associated with them, or have unintentionally crossed geographical barriers to establish elsewhere (*Perrings et al., 2002*; *Perrings et al., 2005*). In Europe, the number of established IAS is estimated between 1,200 and 1,800 species (*DAISIE, 2009*). Annual damage and control costs associated with a set of economically relevant IAS were conservatively estimated at €12 billion for Europe and £1.7 billion for Great Britain (*Kettunen et al., 2008*; *Scalera, 2010*; *Williams et al., 2010*). Moreover, IAS are also a leading cause of biodiversity loss (*Scalera et al., 2012*; *Bellard, Cassey & Blackburn, 2016*). As a result, in line with recommendation of the global Convention on Biological Diversity (CBD, Aichi Target 9; https://www.cbd.int/sp/targets/), policy initiatives are now in place in Europe targeting high profile IAS through trade restrictions, border controls, targeted surveillance as well as early warning, rapid response or management obligations (*Genovesi et al., 2014*; *Tollington et al., 2015*).

Cost-benefit analysis (CBA) is recognised as an important decision support framework for the management of IAS in Europe. Under new European legislation, species identified as posing a high risk will be listed, and Member States will be required to take appropriate action if listed species are found on their territories. This requires a number of processes to identify species, their associated risks and appropriate management options. Species posing high risks are identified based on risk assessments, for which a number of methods have been developed in recent years (*McGeoch et al., 2016*). These have to meet quality standards (*Roy et al., 2014*; *Roy et al., 2017*) and should consider potential damage costs as well as economic benefits of a species. First, when adopting or updating the list of IAS of Union concern (see art. 4 of Regulation (Eu) no 1143/2014 of the European Parliament and of The Council of 22 October 2014 on the prevention and management of the introduction and spread of invasive alien species), the European Commission and Member States need to consider the cost of inaction as well as the cost-effectiveness and socio-economic aspects of listing. Second, derogations from the rapid eradication obligation of regulated species are possible based on either the unavailability of methods, on expected environmental non-target effects of the management measures taken or on a CBA demonstrating with reasonable certainty that the costs will, in the long term, be exceptionally high and disproportionate to the benefits of eradication (*European Union, 2014*). Third, for established IAS of EU concern, Member States are required to put in place effective management measures. Such measures shall be specific to the Member State, be proportionate to the environmental impact and be based on an analysis of the costs and benefits. Cost-benefit analysis including ecological, social and economic aspects is a

prominent requirement of the European IAS regulation. However, it has only rarely been applied in a European context and there are currently no clear standards or guidelines for its application on IAS (*Tollington et al., 2015*).

Given the need for more efficient allocation of scarce conservation resources (*Bottrill et al., 2008*), understanding the costs and benefits of IAS management informs decision making (*Bourdôt et al., 2015*; *Daigneault & Brown, 2013*; *Panzacchi et al., 2007*). When preventive action or early warning mechanisms fail to prevent invasion, eradication is usually considered the preferred option as this avoids future damage costs (*Wittenberg & Cock, 2001*). There are many examples of successful eradications on islands and the mainland (*Robertson et al., 2015b*), yet even with limited invasion extent, the required investment can be considerable (e.g., *Adriaens et al., 2015a*). To assess eradication probabilities, data models based on case studies can be used to underpin decision making on managing IAS (*Drolet et al., 2014*; *Drolet et al., 2015*). Although these models offer interesting tools to guide decisions on IAS management, the lack of published data still prevents their widespread use. If eradication is not feasible, long term control programs can be considered to mitigate IAS impact. The decision to engage in such programs has to consider various aspects to evaluate the feasibility. More recently, invasion scientists and practitioners have focused on developing robust scoring protocols to assess the feasibility of management (*Booy et al., 2017*). These protocols are mostly based on local expert knowledge and consider the species distribution and abundance, the probability of reinvasion, the effectiveness of management options, the cost of management, the non-target impacts of management, the prevailing legislation and a supposed understanding of public attitudes towards the envisaged eradication or management measures. Based on this information, experts then assess the different management options. Such expert elicitation can provide an efficient, transparent tool for decision making (*Burgman et al., 2011*; *Vanderhoeven et al., 2017*). Although management costs are broadly evaluated, the cost of inaction or the cost-benefit ratio of the management strategy are not explicitly considered. Hence, there is a need for decision support frameworks that integrate ecological and socio-economic impacts of IAS with information on the effectiveness and costs of potential management options. CBA offers a framework to combine data on management and damage costs.

Ex ante cost-benefit analysis reveals the management options that yield the highest value for society (*Pearce, Atkinson & Mourato, 2006*; *De Peuter, De Smedt & Bouckaert, 2007*). The management scenarios with the lowest total costs compared with a reference scenario are preferred. For IAS, these costs are typically composed of management costs and damage costs caused by the presence of a species. Benefits accrue over time as increasing damage costs are avoided through management (*Wainger et al., 2010*). The economically preferred management scenario maximizes avoided costs (*Bourdôt et al., 2015*). Accounting for the time value of money, costs and benefits are typically discounted by calculating present values (PV). For management of IAS, the scenario returning the highest net PV (calculated as the total discounted loss prevented through management minus the total discounted implementation cost of management) is preferred. Applying CBA to IAS involves estimating management and damage costs under different population

growth scenarios. Alternative management scenarios are then compared with a business as usual scenario (BAU) which often refers to a scenario where populations are not under coordinated management (*De Wit, Crookes & Van Wilgen, 2001*). Cost-benefit analysis following an established methodology recognizes the real cost of management choices and reveals hidden damage cost and economic benefits (*Pearce, Atkinson & Mourato, 2006*). Performing a CBA however is often data-intensive and examples of comprehensive CBA for IAS are scarce in Europe, but have been produced e.g., for coypu *Myocaster coypus* (*Panzacchi et al., 2007*), common ragweed *Ambrosia artemisiifolia* (*Schou & Jensen, 2017*) and giant hogweed *Heracleum mantegazzianum* (*Rajmis, Thiele & Marggraf, 2016*).

In this study we carried out a CBA for the management of greater Canada goose *Branta canadensis* L. (*Bc*) in Flanders (north Belgium) by additionally performing moult captures on top of hunting and fertility reduction (Fig. 1) using the avoided cost method (*Pearce, Atkinson & Mourato, 2006*). Canada geese have the greatest ecological and economic impact of 26 established alien bird species in the EU (*Kumschick & Nentwig, 2010*). Worldwide, non-native Anseriformes (ducks, geese and swans) mostly have impact through hybridization and herbivory (*Rehfisch, Allan & Austin, 2010*; *Evans, Kumschick & Blackburn, 2016*). Impacts of Canada geese include eutrophication of water bodies, damage to agriculture, animal and human health impacts, damage to recreational areas and an increased risk of birdstrikes (*Maragakis, 2009*; *Van Ham, Genovesi & Scalera, 2013*). The species has already realised most of its potential niche in Europe (*DAISIE, 2009*) including Flanders. Geese can be actively managed through fertility reduction or through culling which involves shooting during the open season for *Bc* and/or capturing flocks of geese during the moult in which they are flightless (*Allan, Kirby & Feare, 1995*). Due to the availability of data on regional population size, economic data on management and damage in the study area and data on the effectiveness of different management strategies for geese in general (*Klok et al., 2010*; *Schekkerman et al., 2000*; *Van der Jeugd et al., 2006*), *Bc* represents a suitable model species for study in a CBA framework. The aim of this paper is to compare a management strategy based on additional coordinated moult captures (hereafter called the enhanced scenario) with a BAU-scenario in which the current active management strategies applying uncoordinated hunting activities and fertility reduction by destroying eggs are continued (*Van Daele et al., 2012*). Non-lethal strategies to mitigate geese impact locally such as discouraging and redistributing geese to alternate foraging sites, scaring, chemical anti-feedants or various forms of habitat management (e.g., *Conover, 1992*; *Melman, De Lange & Clerkx, 2011*) are not considered in this exercise. Although these methods can mitigate damage locally they do not represent population management and are mostly poorly effective in reducing damage as they just shift goose problems to other areas (*Melman, De Lange & Clerkx, 2011*; *Nolet et al., 2016*; *Tombre, Eythórsson & Madsen, 2013*; *Simonsen et al., 2016*). We present a methodology to calculate damage costs associated with eutrophication and damage to cultivated grasslands by *Bc*. We then estimate the management costs under the enhanced scenario, given that both hunting and fertility control which are largely undertaken by non-paid volunteers (hunters and environmental NGO's), are continued. We project the population size over time and calculate the damage costs under both scenarios. Finally, we carry out a sensitivity

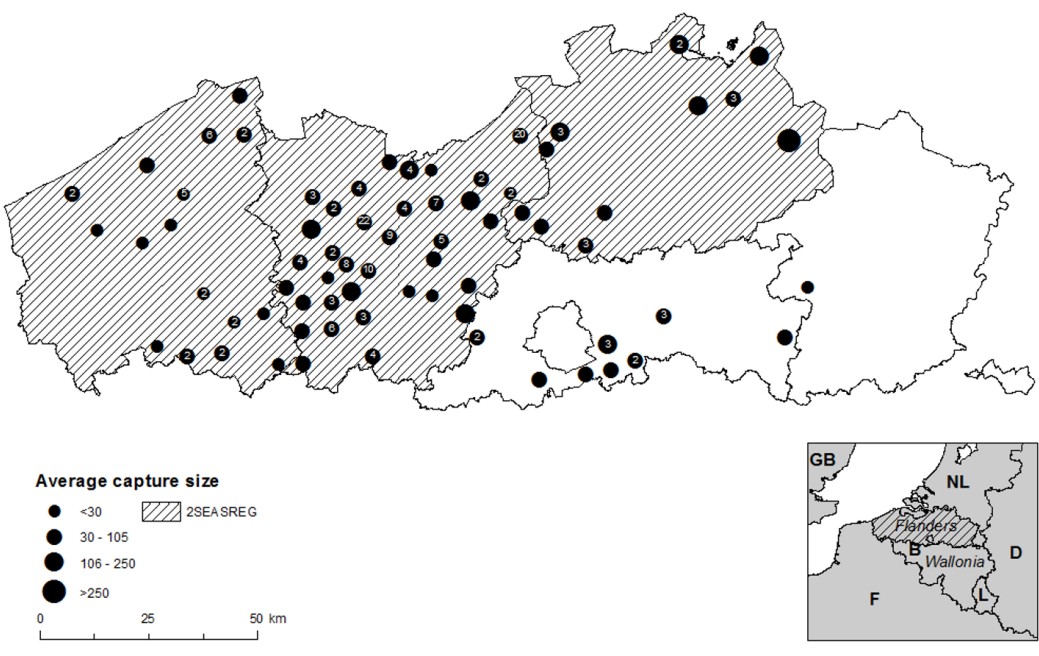

**Figure 1** **Moult capture effort in Flanders (2010–2015).** Moult capture effort (average number of Canada geese captured per municipality) in Flanders (northern Belgium) (2010–2015) with the location of the project area (barred) in northwest Europe: B, Belgium; NL, Netherlands; GB, Great Britain; F, France; D, Germany; L, Luxemburg. Black dots represent average capture size, the number of captures per municipality is shown in the dot.

analysis for population parameters and calculate the difference in PV for a range of possible capture and discount rates. We discuss the strengths and weaknesses of our approach and provide recommendations for the application of this CBA approach to other IAS.

## MATERIAL AND METHODS

We drafted a bio-economic model in a CBA framework (Fig. 2) from the perspective of society in order to minimize the total net social costs associated with *Bc* management in Flanders. We collected information on the biology of the species, its impact and spread, potential management techniques and specific data on the costs of damage. Cost-benefit analysis should include all costs and benefits to all affected parties to reflect the true total impact (*Pearce, Atkinson & Mourato, 2006*). In a conceptual analysis phase, we identified at least six types of impact by *Bc*: eutrophication of water bodies, damage to agricultural crops, birdstrikes, damage to public health and amenities, damage to biodiversity and to recreational areas such as golf courses. Here, we only considered the impact of *Bc* through eutrophication and damage to cultivated grasslands as these forms of damage could be directly or indirectly valued in monetary terms and represent the main economic impacts of *Bc*. Second, we then defined the BAU scenario as the current management practice with comprises uncoordinated shooting and fertility reduction and the enhanced scenario which supplements the BAU scenario with moult capture. We then collected data on the costs of
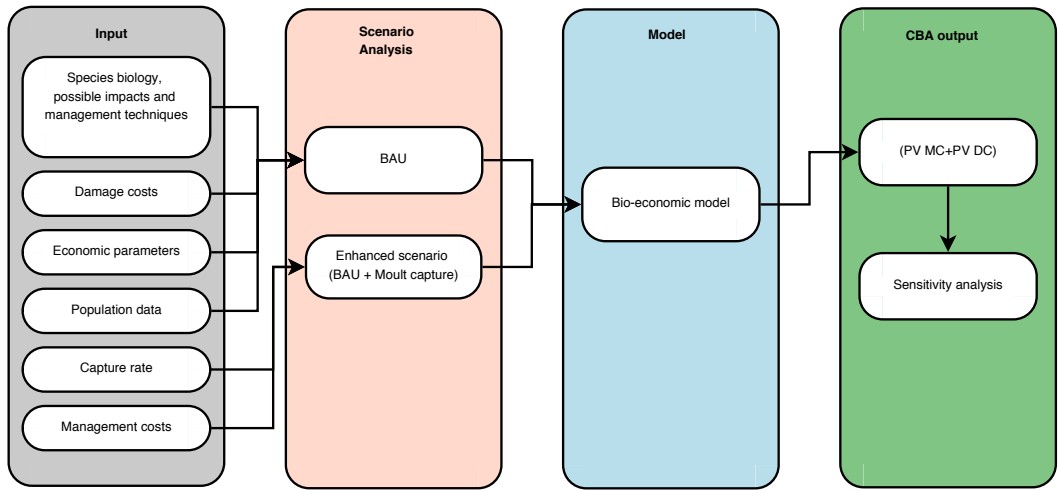

**Figure 2** **Cost-benefit analysis framework.** Schematic representation of the cost-benefit analysis framework for greater Canada goose *Branta canadensis* L. in Flanders (Northern Belgium).

control of the *Bc* population from the principal stakeholder in goose management in the project region. We derived a realistic capture rate based on management and *Bc* population data over recent years. Then, the population was modeled using a logistic growth curve based on 1992–2009 census data. Subsequently, the *Bc* population was projected to the year 2050 under the two scenarios. Thereafter, the data was combined in a bio-economic model. The time horizon for our CBA is the period 2016–2050. For each scenario we calculated the sum of the present value of management and damage costs. Finally, we conducted a sensitivity analysis on several parameters to test the robustness of our results. First, we varied the population parameter for the intrinsic growth rate $r$ and the carrying capacity $K$ of our logistic growth model. Second, we reduced the sales price of hay by 90% to test whether BAU would become preferable in a scenario with almost no agricultural damage. Third, we varied the capture rate as this parameter directly influences the total management cost. Finally, we also varied the discount rate.

## Project area and target species

Belgium is a federal country with three administrative regions (Flanders, Wallonia and the Brussels Capital Region) each with their own regional government. Flanders (13,522 km$^2$) is highly urbanized with a landscape consisting of a fragmented and complex mosaic of different forms of land use, primarily agricultural areas (45%), built-up land (26%), areas protected under different nature conservation legislations (8%) and other infrastructure (*Poelmans & Van Rompaey, 2009*; *Adriaens et al., 2015b*). In Flanders, several populations of geese have impact on biodiversity and society, including invasive non-native *Bc*, native greylag goose *Anser anser*, feral domestic goose *A. anser* f. domestica, mixed populations of wild and domesticated barnacle goose *Branta leucopsis*, as well as a number of non-native species like Egyptian goose *Alopochen aegyptiacus*, bar-headed goose *A. indicus* and upland goose *Chloephaga picta* (*Vermeersch, Anselin & Devos, 2006*). Of these, *Bc* (11,000 birds),

greylag goose (19,000) and Egyptian goose (3,000) are the most abundant species. As count coverage is not complete, these numbers are probably an underestimation of real population numbers present (*Devos & Onkelinx, 2013*). Data on compensation payments in the period 2009–2011 show *Bc* is the most important goose species causing agricultural damage in Flanders in terms of compensation payments to farmers as well as in diversity of crop damage (*Van Gils et al., 2009*; *Van Daele et al., 2012*). Canada geese started breeding in the wild in 1973 but have increased since the nineties to about 1,800 breeding pairs in 2000–2002 (*Vermeersch et al., 2004*). Based on winter census data, the post-breeding population stabilized with an average winter maximum of 11,359 *Bc* in the period 2010–2015 (*Devos & Onkelinx, 2013*; *Devos & T'Jollyn, 2016*). Impacts of *Bc* in Flanders include crop damage, eutrophication of ponds and fens, overgrazing, fouling and trampling of vegetations such as reed beds and meadows, soil and water pollution, pathogen transmission and hybridization with native species. Several case studies in Flanders show the presence of *Bc* hampers costly nature restoration projects because of nutrient enrichment through their faeces (*Van Ham, Genovesi & Scalera, 2013*). Based on ringing data, *Bc* can undertake long-distance dispersal within northwest Europe (*Voslamber, 2011*), but the population in Flanders is considered relatively sedentary with birds primarily moving locally for foraging, breeding and moulting, and their home ranges seldom exceed a 50 km radius (*Cooleman et al., 2005*). To reduce their impact, *Bc* are managed in Flanders in an adaptive management approach, using an integrated strategy which involves hunting (*Bc* is a game species), fertility reduction (egg pricking) and moult capturing which has been upscaled and intensified in recent years. *Bc* are highly susceptible to moult captures with considerable numbers being caught yearly (on average 2,000 *Bc* per year in the period 2009–2012). *Bc* represent 87% of the geese caught in such captures (*Van Daele et al., 2012*). Summer census of the population has shown a significant decrease in *Bc* numbers since 2010 (*Huysentruyt et al., 2013*; *Adriaens et al., 2014*). Because wildlife management in Belgium is a responsibility of the regions, *Bc* show limited dispersal, data consistency and data quality is good for Flanders and this is where most management is currently undertaken, the geographic scope of this CBA is the Flemish region only.

## Calculation of damage costs

Greater Canada geese are known to exert severe pressure on small water bodies such as ponds, reducing water quality through eutrophication (*Allan, Kirby & Feare, 1995*; *Gosser, Conover & Mesmer, 1997*; *Kumschick & Nentwig, 2010*; *Smith, Craven & Curtis, 2000*). This involves the deposition of high nutrient loads, notably nitrogen (N) and phosphorous (P) (*Smith, Tilman & Nekola, 1999*). The total nutrient input of *Bc* in the environment was calculated based on Canada geese producing about 500 g of droppings per day with a moisture content of 80% and nutrient load concentrations for N and P of 24.2 mg/g and 3.6 mg/g of dry matter respectively (*Ayers et al., 2010*; *Van Daele et al., 2012*). Damage costs for N and P were valued in the range of 5 €–74 €/kg and 80 €–800 €/kg (2010 prices) respectively based on the Flemish environmental cost model for water sanitation (*De Nocker, Broekx & Liekens, 2011*; *Liekens et al., 2013*). As such, we use the cost of water sanitation as a proxy to calculate damage through eutrophication. We calculated total

damage costs for eutrophication, multiplying the estimated number of geese per year by an estimated damage cost per goose. Damage costs were calculated under two scenarios, assuming the lowest and the highest unit cost values for N and P respectively. Since most water sanitation techniques reduce both N and P simultaneously, we did not consider the maximum value for both nutrients simultaneously as this would overestimate the true damage cost (*Liekens et al., 2013*). Nitrogen concentration in geese droppings is much higher than the phosphorous concentration. In the "high" variant of the two scenarios, we therefore used the highest unit cost value for N and the lowest for P removal respectively.

Canada geese damage crops by foraging and trampling on agricultural fields (*Van der Jeugd et al., 2006*). In the Netherlands, 58%–80% of compensation payments to farmers were made for damage to grasslands by foraging geese (*Lemaire & Wiersma, 2011*). Geese in Flanders spend about 90% of their time on grasslands (*Huysentruyt & Casaer, 2010*; *Van Gils et al., 2009*). Also, winter wheat is a crop often affected by *Bc* (*Van Gils et al., 2009*). Data on agricultural damage costs in relation to *Bc* numbers were lacking for Flanders. We therefore relied on data from the Netherlands (Data S1). We used seasonal data on compensation payments and geese numbers for greylag goose (*Lemaire & Wiersma, 2011*). This species is abundant in the Netherlands and has similar feeding habits. However, we applied a correction factor of 1.26 to account for the higher daily energy intake of *Bc* compared to *A. anser* (*Lemaire & Wiersma, 2011*). We used greylag geese numbers in January as a proxy for the year round geese population because counts of geese were most complete for that month (H Schekkerman, pers. comm., 2015). We further summed the total damaged area over the different seasons. We drafted a regression model on this yearly dataset for the total damaged area and the number of geese assuming all damage could be attributed to cultivated grassland at a yield of 10 tonnes hectare$^{-1}$ year$^{-1}$ (*Zwaenepoel, 2000*). This type of grassland is the most prevalent in Flanders (*Demolder et al., 2014*; *Wils et al., 2006*). The total area of crop loss by *Bc* was then estimated applying the resulting model to the estimated *Bc* population for Flanders. We valued yield loss using a 2014 sales price of hay of 0.12 €/kg as published by the Belgian *Federal Public Service Economy (2015)*. This price represents an average sales price the farmer can get and is based on *Eurostat (2008)*. We use this price, which does not include subsidies or taxes, as a proxy, as true market prices are unlikely to reflect the true social value of a resource.

## Calculation of management cost for moult capture

Management costs for moult capture were based on data provided by RATO vzw, the principal organisation undertaking moult captures of *Bc* in Flanders. We calculated representative costs for a capture of flock sizes ranging between [30, 105] geese (small capture) and between [105, 205] geese (large capture) including the costs of preparation (prospecting, planning, requests for permission and permits), transport, personnel and materials used. A small capture involved a cost of 1,005 €, a large capture 1,253 € (Table 1). Note there is no big difference in cost between the two capture sizes. Therefore, we assumed the costs of maintaining a constant capture rate were constant over a range of population densities and thus do not vary within the two capture sizes. Geese naturally flock together on a limited number of suitable moulting sites that are well known to the manager and

**Table 1  Capture size, rate and cost.** Percentage and average number of geese captured in small and large capture events and their associated cost based on data from goose captures in Flanders (period 2010–2014).

| Capture size | % of captured birds | Average number captured | Calculated representative cost per capture (€) |
| --- | --- | --- | --- |
| Small [30, 105] | 51% | 46 | 1,004.93 |
| Large [105, 205] | 49% | 122 | 1,253.15 |

every capture requires a minimum number of staff. The difference in costs is mainly due to an increase handling time for larger captures and the use of extra vehicles which results in higher transportation costs. Consequently, the average costs per goose were lower as the number of captured geese increased.

As the costs for moult capture mainly depend on the number of captures and not on the number of geese, we estimated the number of captures needed to reduce the *Bc* population by 50% per year applying three steps. First, we calculated the total number of geese captured per year based on the 50% capture rate. Second, based on real data of capture events from Flanders in the period 2010–2014, we estimated the percentage of the total number of *Bc* captured in either a large or small capture event (Table 1). Third, we calculated the average number of *Bc* captured for each of the two categories based on the same data. We then applied these percentages to the total captured population per year to distribute the yearly number of geese captured over the two capture types (large or small). Dividing this number by the average number of geese captured per capture type in the period 2010–2014 determines the number of captures needed. To calculate the moult capture cost, we multiplied this number of captures by the cost per capture. Finally, we calculated total management cost (2014 prices) by multiplying the number of captures for each capture size with the corresponding cost per capture for that capture size.

## Population model

Under both scenarios (BAU and enhanced scenario) we assumed the growth of the *Bc* population could be described by a logistic growth model (*Trost & Malecki, 1985*) as shown in (1) where $K$ is the carrying capacity, $A$ equals $\frac{K-P(0)}{P(0)}$, $t$ is the time, $P(0)$ the initial population at $t=0$ and $r$ the intrinsic growth rate (*Tsoularis & Wallace, 2002*).

$$P(t) = \frac{K}{1 + Ae^{-rt}}. \tag{1}$$

The annual population size of *Bc* was taken from the Flemish waterbird census for the period 1992–2014 (*Devos & Onkelinx, 2013*; Data S2). Because geese have been systematically captured since 2010 (Fig. 1) our population parameter estimates would be biased if we included the post-2009 years in the analysis. We therefore limited the dataset to the period 1992–2009. Non-linear least squares regression (NLS) (*Montgomery, Peck & Vining, 2012*) was applied to fit the model to the data. Using multiple starting values for $r$ and $K$ we tested if the algorithm converged to the same parameters in each estimation. We defined ranges of [0.25−3] and [5.000, 30.000] for $r$ and $K$ respectively and uniformly divided these into 10 pairs of $r$ and $K$ starting values. We then re-estimated the model for each pair of starting values (Table 2).

**Table 2  Parameter estimates logistic growth curve.** Parameter estimates for the logistic growth curve at ten different pairs of starting values for $K$ (carrying capacity) and $r$ (intrinsic growth rate).

| Startvalue $K$ | Startvalue $r$ | $\hat{K}$ | $\hat{r}$ | $se(K)$ | $se(r)$ |
|---|---|---|---|---|---|
| 5000.0000 | 0.2500 | 10753.5900* | 0.4838* | 408.8097 | 0.0142 |
| 7777.7780 | 0.5556 | 10753.6000* | 0.4838* | 408.8102 | 0.0142 |
| 10555.5560 | 0.8611 | 10753.6000* | 0.4838* | 408.8100 | 0.0142 |
| 13333.3330 | 1.1667 | 10753.6000* | 0.4838* | 408.8100 | 0.0142 |
| 16111.1110 | 1.4722 | 10753.6000* | 0.4838* | 408.8101 | 0.0142 |
| 18888.8890 | 1.7778 | 10753.6000* | 0.4838* | 408.8101 | 0.0142 |
| 21666.6670 | 2.0833 | 10753.6000* | 0.4838* | 408.8107 | 0.0142 |
| 24444.4440 | 2.3889 | 10753.5900* | 0.4838* | 408.8094 | 0.0142 |
| 27222.2220 | 2.6944 | 10753.6000* | 0.4838* | 408.8100 | 0.0142 |
| 30000.0000 | 3.0000 | 10753.6000* | 0.4838* | 408.8101 | 0.0142 |

**Notes.**
*Significant at $p < 0.01$.
$r$, intrinsic growth rate; $K$, carrying capacity.

## Capture rate

The capture rate was defined as the ratio of captured geese divided by the sum of captured and counted geese after the moult capture season. To assess this capture rate, we used data from *Van Daele et al. (2012)* for Flanders. Estimates based on these data range from 41% to 56%. A 50% capture rate in the enhanced scenario therefore seemed a reasonable value. We further supposed a reduction in geese numbers by moult captures would not affect the parameters of our population model and assumed immigration and emigration to be zero. As the population reproduces before moult capturing, we model the population growth realized after the previous moult capture in a given year (post-moult capture population) and before the next moult capture one year ahead (pre-moult capture population). As we assume a constant capture rate, the post-moult capture population is known. Thus, we can compute equation (2) (the inverse of equation (1)) yielding a value for time $t$ which corresponds to the same population level as the post-moult capture population (Fig. 3). This way, we can compute the *Bc* population one year ahead.

$$t = -\frac{1}{r}\ln\left(\frac{K - P(t)}{AP(t)}\right). \tag{2}$$

## Present value and sensitivity analysis

We combined all data in a bio-economic model to simulate management and damage costs for the BAU-scenario and the enhanced scenario. We used the PV to compare the two scenarios.

$$PV = \sum_t \frac{M_t + D_t}{(1+i)^t}. \tag{3}$$

The formula for the PV is shown in (3) where $M_t$ is the management cost at time $t$, $D_t$ damage cost at time $t$ and $i$ is the real discount rate. We thus calculate the total discounted

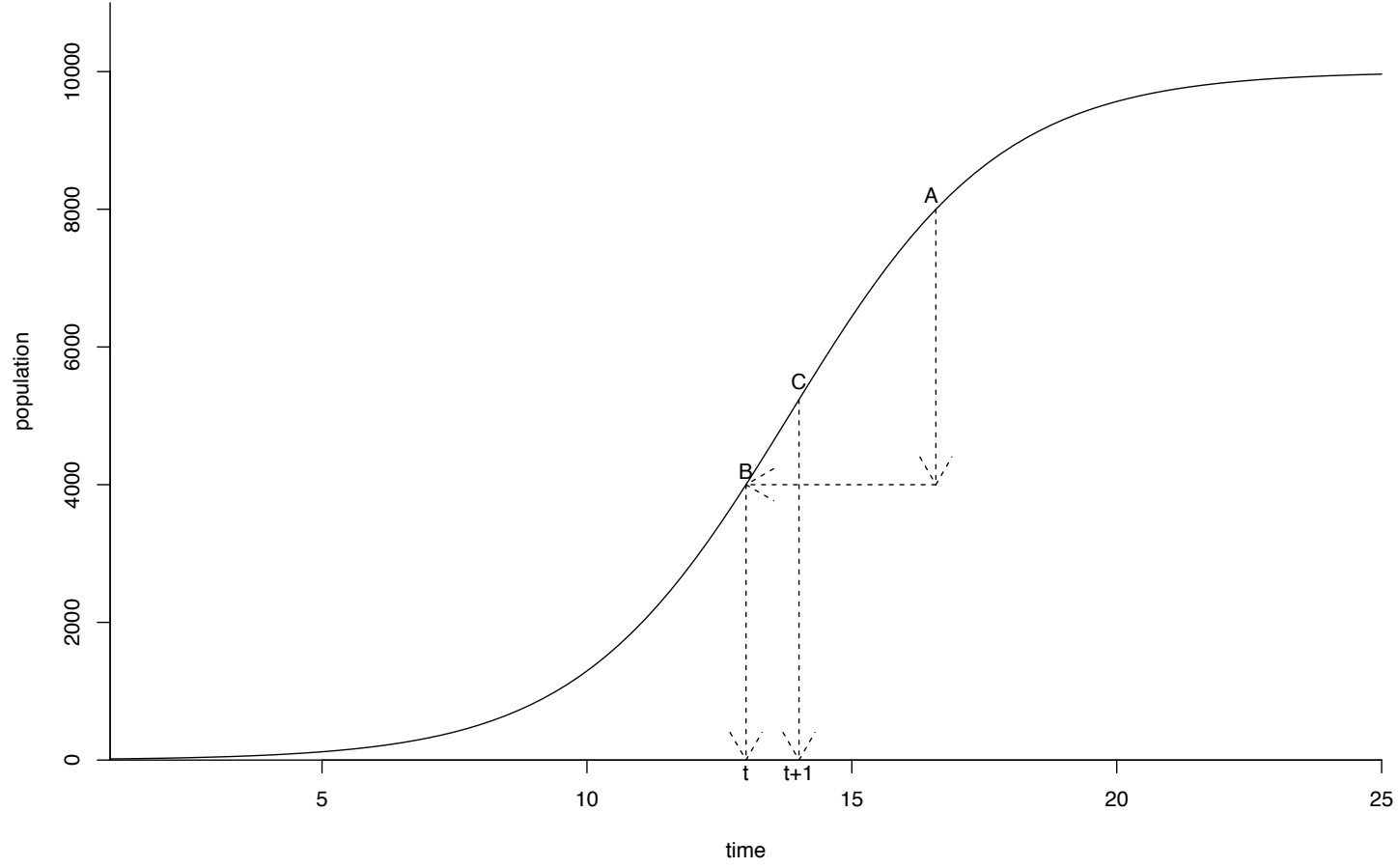

**Figure 3 Schematic representation of the modelled *Branta canadensis* population growth between two successive moult captures.** (A) is the pre-moult capture population in a given year, (B) represents the post-moult capture population in the same year (pushing the population down on the logistic growth curve). (C) is the pre-moult population on the next year. The *X*-axis represents a time index.

costs under the two scenarios. Discounting costs and benefits is common practice in CBA (*Bourdôt et al., 2015*; *Daigneault & Brown, 2013*; *Pearce, Atkinson & Mourato, 2006*). The yearly discount rate was set at 4% based on guidelines for valuing ecosystem services (*Liekens et al., 2013*). We discounted management and damage costs to the year 2015 using constant prices of the year 2014. To update unit costs for eutrophication to the 2014 price level, we followed *Liekens et al. (2013)* applying the historical yearly average consumer price index (CPI) with the base year 2004 (National Bank of Belgium). Management costs and damage costs for lost harvest were already expressed in the 2014 price level.

Sensitivity analysis was carried out by varying the values for the observed capture rate (41% and 56%) and the discount rate, to assess the change of the PV. First, we varied the capture rate by a 10% decrease and increase respectively. These simulations represent a situation with a slower (capture rate—low scenario) and faster (capture rate—high scenario) reduction of the population by moult capture than the observed values respectively. Second, we varied the real discount rate from the initial 4% to 2.5% as suggested by *Liekens et al. (2013)* and *Perman et al. (2003)*. Third, two scenarios were

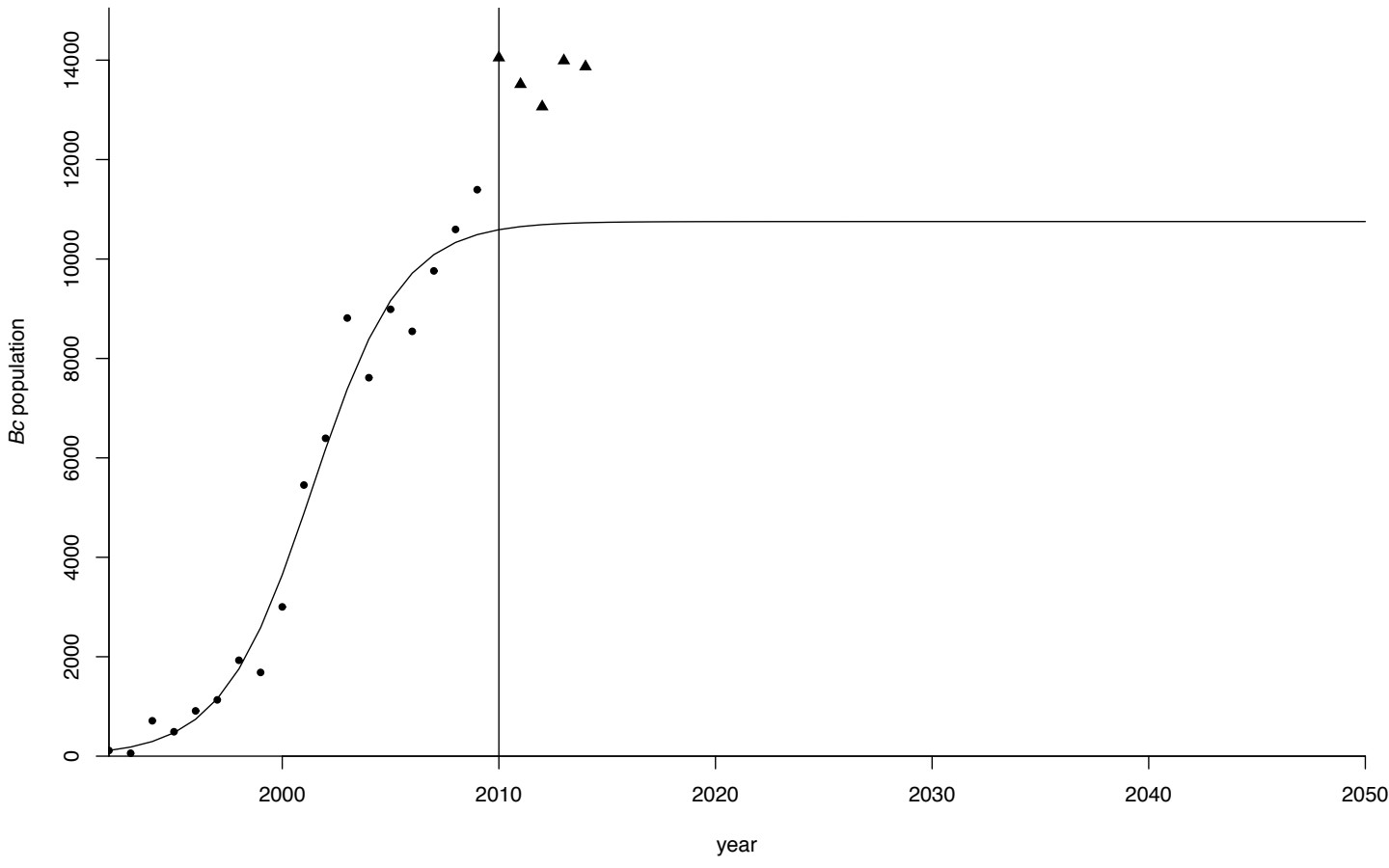

**Figure 4** **Logistic growth curve for greater Canada goose in Flanders.** Projection (dots) of the greater Canada goose (*Branta canadensis*) population in Flanders until 2050 under a logistic growth curve. Observed values post-2009 are plotted as triangles.

calculated in which we increased the population parameters $r$ and $K$ by 10%. We changed either $r$ or $K$ but not both at the same time. Finally we reduced the sales price of hay by 90%.

## RESULTS

### Population model
The estimates for $r$ and $K$ in the population model using the different starting values converged to the same values in all regressions, indicating the robustness of the estimates (Table 2, Fig. 4). Both parameter estimates for $r$ and $K$ were significant at $p < 0.01$ in all regressions. The estimate for the carrying capacity (10,753 birds) was consistent with *Van Daele et al. (2012)* who indicated geese numbers stabilized at a population of 10,000−12,000 birds. In our model, the population reached this level in 2010.

### Present value
The bio-economic model output showed PV was about nine times lower under the enhanced scenario compared to the BAU scenario. The pooled linear regression model for the estimation of agricultural damage fitted the data well with the number of geese
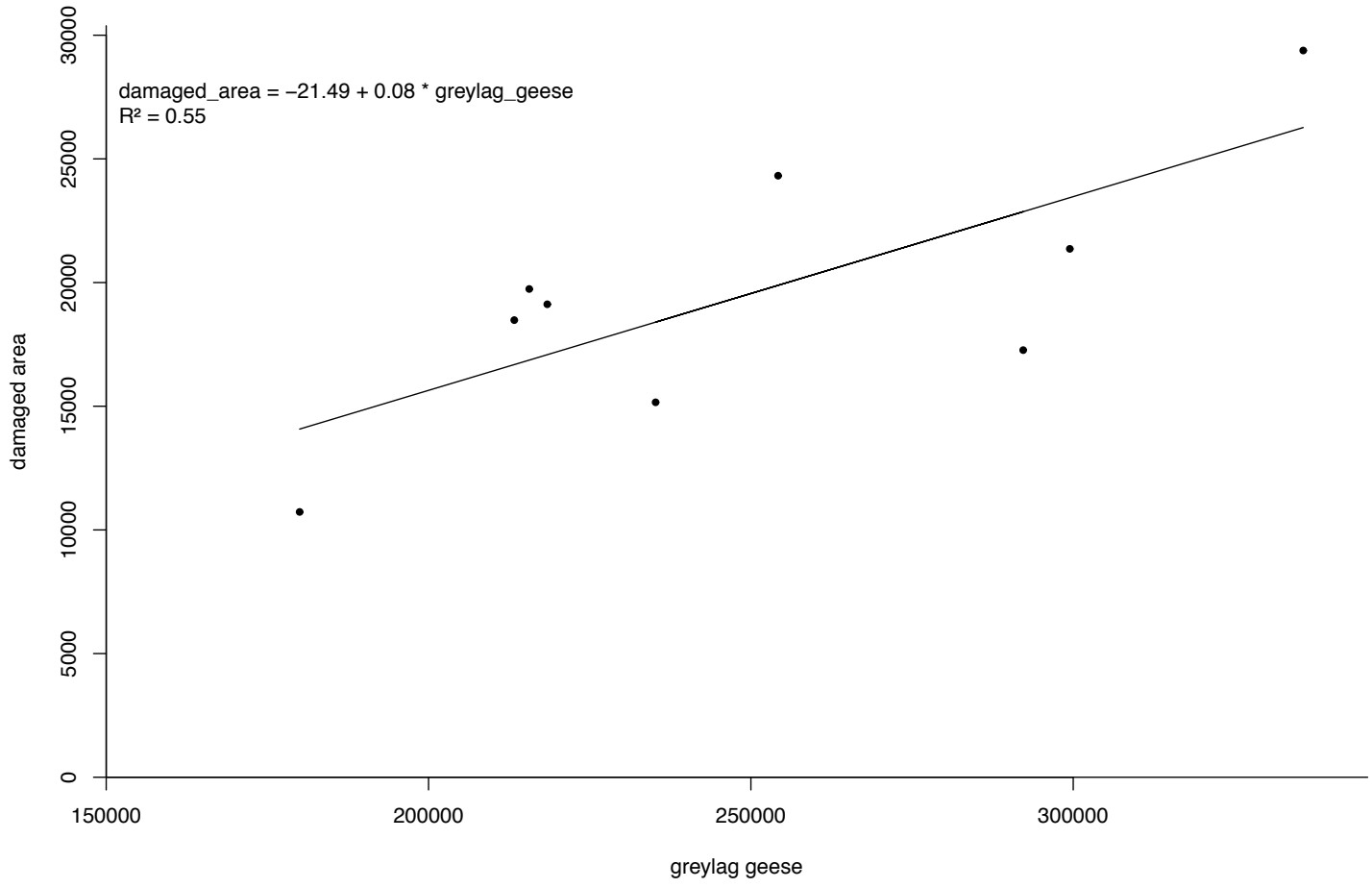

**Figure 5** **Damage density curve greylag geese.** Simple linear regression model of the number of greylag geese *Anser anser* in the Netherlands versus damaged area.

accounting for 55% of the variation in the damaged area (Fig. 5). As such, the agricultural losses avoided in the period 2016–2050 under the enhanced scenario amount to an estimated 21,700 k€. For eutrophication, the avoided damage ranged from 2,920 k€ to 14,850 k€ depending on the unit costs for eutrophication applied. Depending on the unit costs for eutrophication, we found a difference in PV for the BAU and enhanced scenario of 24,370 k€ and 36,300 k€ (Table 3).

## Sensitivity analysis

Applying different values for the capture rate or the discount rate in the model did not influence the general outcome. At a lower than observed capture rate of 36.9% (10% lower as the lower bound of 41% of the observed rates) as opposed to a 50% capture rate, *Bc* could not be eradicated within the time horizon 2016–2050. Management costs increased with 120 k€ in that scenario when compared with the base scenario of the enhanced scenario (Table 3). The PV under this scenario was still four times lower than the BAU scenario, indicating performing additional moult captures was still preferable over BAU. Increasing

**Table 3  Present value calculations.** Present value (PV, M€) calculations for the management cost (MC) and damage costs (DC) for the base scenarios (BAU and enhanced scenario) and the low and high capture rate scenarios used in the sensitivity analysis.

| Type of cost | Base scenarios | | | Capture rate low | | Capture rate high | | Discount rate = 2.5% | | |
|---|---|---|---|---|---|---|---|---|---|---|
| | BAU | Enhanced | Δ PV | Enhanced | Δ PV | Enhanced | Δ PV | BAU | Enhanced | Δ PV |
| **PV Damage Costs (DC)** | | | | | | | | | | |
| Agriculture | 24.05 | 2.35 | 21.70 | 5.09 | 18.97 | 1.50 | 22.55 | 29.83 | 2.47 | 27.36 |
| Eutrophication | | | | | | | | | | |
| Unit cost of N and P (low)[a] | 3.24 | 0.32 | 2.92 | 0.69 | 2.55 | 0.20 | 3.04 | 4.02 | 0.33 | 3.68 |
| Unit cost of N and P (high, low)[a] | 16.46 | 1.61 | 14.85 | 3.48 | 12.98 | 1.03 | 15.43 | 20.41 | 1.69 | 18.72 |
| **PV Management Costs (MC)** | 0.00 | 0.25 | −0.25 | 0.37 | −0.37 | 0.22 | −0.22 | 0.00 | 0.27 | −0.27 |
| **PV DC + PV MC** | | | | | | | | | | |
| Eutrophication | | | | | | | | | | |
| Unit cost of N and P (low) | 27.29 | 2.93 | 24.37 | 6.14 | 21.15 | 1.92 | 25.37 | 33.85 | 3.07 | 30.78 |
| Unit cost of N and P (high) | 40.52 | 4.22 | 36.30 | 8.94 | 31.58 | 2.74 | 37.77 | 50.24 | 4.43 | 45.82 |

**Notes.**
[a]High–low unit costs (2014 prices) for N: 5.4–79.94 /kg and low for P: 86.42/kg

the capture rate to 61.6% (10% higher as the upper bound of 56% of the observed rates) decreased management costs by 30 k€ when compared to the base scenario of the enhanced scenario. Here, the population reduced faster resulting in a smaller number of captures required. Model output showed that with the higher capture rate the PV was about 14 times lower under the enhanced scenario compared to the BAU scenario. Under the BAU scenario yearly damage costs rapidly become constant since geese numbers stabilized. Under the enhanced scenario damage costs declined over time. If the discount rate dropped from 4% to 2.5%, the difference between BAU and moult capture scenario increases due to a higher discount factor $(1/(1+i)^t)$. Note that in this scenario the BAU costs also change since management and damage costs in both scenarios are similarly discounted with the same discount rate. In the enhanced scenario the ratio of the PV increased to 11 as opposed to 9 in the base scenario. With population parameters $r$ and $K$ increased by 10% the enhanced scenario is preferred over the BAU-scenario. Even in case of a drastic reduction of 90% in the sales price of hay the total damage costs were still higher than the management costs at the lowest unit prices for N and P (Table S1). Clearly, the management costs in general are low in all enhanced scenarios.

# DISCUSSION

The EU-regulation 1143/2014 on the prevention of spread and introductions of IAS requires Member States to conduct cost-benefit analysis in order to identify cost effective control measures to minimize and mitigate IAS impacts. However, performing CBA is often not straightforward since it requires a lot of data on all costs and benefits as well as clear guidelines to decide on underlying assumptions. The relative complexity of CBA in
comparison to other methods (e.g., effectiveness analysis, multi criteria analysis) renders the method less useful in support of derogations on the rapid response obligation. However, CBA is especially useful for decision making on the management options for established IAS as it allows assessment of the real management and damage costs under different management scenarios, including the zero management option.

Cost-benefit analysis for *Bc* in Flanders shows that complementing the current management actions with coordinated moult captures significantly reduces damage costs associated with eutrophication and agricultural losses. Our approach almost certainly underestimated the costs of damage. Although we considered two major impacts (eutrophication and damage to cultivated grasslands), these only represent two of the six impact types identified. In practice, the included costs and benefits in a CBA are often limited to those that are measurable (*Weatherly et al., 2009*). In a conceptual analysis phase, we identified at least six types of impact by *Bc*: eutrophication of water bodies, damage to agricultural crops, birdstrikes, damage to public health and amenities, damage to biodiversity and to recreational areas such as golf courses. Several of those impacts were not taken into account in the model for various reasons. First, for some impacts assessing their magnitude is complex. For example, *Bc* can have an impact on biodiversity by competing with other bird species (*Kumschick & Nentwig, 2010*; *Rehfisch, Allan & Austin, 2010*) although this is seldom quantified and has been challenged by other authors (*Strubbe, Shwartz & Chiron, 2011*). Also, *Bc* is known as an opportunistic species, which breeds early in season and can easily colonize new nesting sites at the expense of other waterfowl (*Titchenell & Lynch, 2010*). Canada geese can destroy conservation value habitat by trampling, leaving impoverished habitat to other wildlife but this effect is often context dependent and difficult to assess (*French & Parkhurst, 2009*). Modelling interactions between *Bc* and other species is also complex and difficult to value.

Second, although some impacts are quantifiable, data were lacking on the magnitude and extent to which they occur in the study area. For instance, aviation safety is a federal matter so information on the number of birdstrikes only exist for Belgium as a whole and not at the regional level of the study area. *Bc* are recognised as a high risk species for birdstrike, where their large body size and flocking behaviour increase the risk of multiple damaging strikes (*Maragakis, 2009*). In addition to the infrequent costs of catastrophic damage, birdstrikes bring significant costs through increased repairs and delays. This cost was not taken into account in this study but is significant in other countries (*Allan, 2002*). However, some of the effects we did not take into account are expected to be rather small. The effect of geese on golf courses was discarded after a rough calculation of the damage costs. Considering the number of clubs on the web page of the Flemish Golf Association (54 clubs) and crude data on the estimated damage cost per club of which 20% can be attributed to *Bc* (*Williams et al., 2010*), we derived a total damage cost of 60 k€. However, this calculation was not based on actual geese numbers on golf courses and we therefore did not include it in the analysis. We also expect the damage to public health through direct contact of humans with *Bc* to be insignificant. Although *Bc* are susceptible to highly pathogenic avian influenza (*Pasick et al., 2007*), transmission of disease or parasites from geese to humans has not been well documented, and human health impact would rather occur indirectly

through contact (swimming) with contaminated water or goose droppings (*Converse et al., 1999*; *Fallacara et al., 2001*). Therefore, the choice to consider eutrophication and loss to agricultural crops was a pragmatic approach based on available data. We assumed these two impacts represent the highest share in total damage incurred by *Bc*.

Also, within the agricultural damage considered, we only included part of the potential economic cost i.e., damage to cultivated grasslands. Although we know from empirical data this represents the predominant proportion of crop damage, other type of crops (e.g., winter wheat) are also affected (*Van Gils et al., 2009*). Estimating the total damage on all crops requires detailed understanding of the foraging behaviour of *Bc*, their distribution and abundance in relation to the different crop types. Additionally, a register of damage to crops including affected area and/or compensations paid to farmers would be needed. Currently, these data do not exist for *Bc* in Flanders. *Bc* is a game species and only damage of *Bc* originating from nature reserves are eligible for damage compensation by the government. Moreover, the minimum damage cost has to be 300 €, of which 250 € is considered to be the risk to be covered by the farmer himself, and farmers have to show they applied preventive measures and have to report damage in a timely manner. Alternatively, data could be collected at a sample of farms through detailed monitoring of geese numbers and the area damaged. Using productivity estimates of agricultural land, the value of the total crop loss could then be calculated using average sales prices (*Eurostat, 2008*). Extrapolation could then be used to assess the total damage for the study region. Since data were lacking we based our estimates on data for another species, greylag goose, applying a correction factor for the higher energy intake by *Bc*. Although *Bc* and greylag goose have comparable feeding ecologies and predominantly feed on grasslands, collecting real data on the extent of damage by *Bc* is recommended for two reasons. First, the data could validate the current approach. Second, the real data for *Bc* collected could directly be applied in a damage assessment if enough observations were available for a robust estimation. We believe the results of our CBA are robust since extending the scope of the damage costs would render the enhanced scenario even more preferable.

Our CBA approach considered management cost and damage costs but did not consider other type of values associated with *Bc* e.g., ornamental value, value as a game species, meat production, ecosystem services associated with the species. Existence values or recreational values are components of the total economic value but were not estimated in our cost-benefit framework. Although valuation methods to estimate the magnitude of these type of values exist in the field of environmental economics (*MacMillan, Hanley & Daw, 2004*), they are generally difficult to quantify. Such methods include contingent valuation to estimate willingness to pay to approximate existence values and the travel cost method to assess the recreational value (*MacMillan, Hanley & Daw, 2004*; *Pearce, Atkinson & Mourato, 2006*). Other studies have addressed this issue applying benefit transfer (*Plummer, 2009*) or using stated preference techniques (*Rajmis, Thiele & Marggraf, 2016*). However, benefits incurred in one region are not necessarily transferable to other study areas. We could not find specific studies for Flanders in which existence values or recreational values for *Bc* are provided. If available, such data would render our CBA more realistic.

Conducting a CBA generally requires a set of assumptions. First, the timeframe for which costs and benefits are calculated has to be determined. According to *Pearce, Atkinson & Mourato (2006)* there are no clear-cut rules to choose a reasonable period. *Emerton & Howard (2008)* argued in favour of a ''sufficiently large'' timeframe, in order to capture all potential impacts. Here, we chose to project the goose population until 2050. Census data, particularly post-breeding counts of the wintering population, show *Bc* numbers are stabilizing in Flanders, indicating the population is close to carrying capacity (*Devos & Onkelinx, 2013*), a conclusion supported by our analysis. We therefore think the selected time period was large enough for our purpose. Also, at an assumed capture rate of 50%, our model indicates *Bc* could be eradicated before that date. However, in reality, an assumed capture rate does not consider real world operational problems with which managers are confronted e.g., the increase in searching costs when the species is getting scarcer. *Smith, Henderson & Robertson (2005)* also assumed the same effort was required to reduce a duck population by 50%, regardless of the number of animals involved. The predictions from the *Smith, Henderson & Robertson (2005)* model were close to the observed results of a subsequent eradication (*Robertson et al., 2015a*). Therefore, an assumed constant capture rate is not an unreasonable simplification.

Second, as the population model represents a key component of our bio-economic model, we required another set of assumptions. Population losses through other methods than moult capture, such as fertility reduction and shooting during the open season for *Bc*, were considered to stay proportional to changes in population sizes due to moult captures.

Fertility control by egg reduction was thought to have only a minor impact at the population level, unless conducted in a coordinated manner and over long periods of time alongside other lethal control (*Klok et al., 2010*). In Flanders, the rather small effect of fertility control at the regional scale reflected the spread and limited accessibility of nests. Yet, the method is frequently used on a local scale (municipality ponds, small nature conservation areas, recreational areas), to lower goose numbers during spring and limit local grazing and eutrophication impacts. Reported numbers of *Bc* culled by shooting have shown a proportional increase with *Bc* numbers in Flanders (*Adriaens et al., 2012*; *Scheppers & Casaer, 2008*). Therefore, in this CBA we consider moult capture as an additional management action that supplements the BAU scenario and assume that the relative contribution of other management measures to population development remain constant under the enhanced scenario. Population modelling has shown culling birds is more effective in reducing bird numbers than egg reduction irrespective of density dependence (*Klok et al., 2010*). *Bc* is a game species in Flanders and good numbers are harvested yearly during the open season.

Also, with good time series of goose counts, we assumed a logistic growth curve for the population to estimate intrinsic growth rate and carrying capacity. While a matrix population model, considering reproduction and survival at different life stages, might more accurately project population numbers (*Caswell, 2001*; *De Kroon et al., 1986*), these models require detailed data on population parameters related to survival, growth rates and fertility of different life stages (*Klok et al., 2010*). Such data are currently not available for the Flemish population. Considering the need for long time series of goose counts to

inform the population model and to estimate carrying capacity, the methodology cannot be usefully applied to newly introduced non-native species. For such species, models could rely on species distribution modelling to estimate the carrying capacity (e.g., *Strubbe & Matthysen, 2009*), and data on their intrinsic growth rates.

Another parameter often discussed in literature is the discount factor $1/(1+i)^t$. The discount factor has the consequence that future costs and benefits have less weight in the analysis. As with many environmental and biodiversity related investments, benefits (avoided damages) become apparent only after some time while the costs occur earlier in time. Therefore, the benefits could be undervalued and costs overestimated. From a societal perspective with sustainability becoming increasingly important in economic decision-making, a high discount rate could rapidly make future costs and benefits insignificant thereby impacting future generations (*Pearce, Atkinson & Mourato, 2006*; *Scarborough, 2011*). For the discount factor, we relied on recommended reference values available in a regional guideline (*Liekens et al., 2013*). *Perman et al. (2003)* also note a discount rate varies between 2 and 5%, with a recommendation to use a real discount rate of 4% in CBA.

Finally, we also assumed the benefits of management under an enhanced scenario were not offset by potential increases in the abundance of other goose species such as greylag goose *A. anser* or feral goose. As these species exhibit similar habitat and feeding characteristics (*Huysentruyt & Casaer, 2010*; *Lemaire & Wiersma, 2011*), lowering *Bc* numbers through management could release them from interspecific competition which could offset some of the benefits of *Bc* management e.g., through increased agricultural damage. To include such multi-species effects in modelling requires further detailed monitoring of geese populations in the study area.

Cost-benefit analysis can inform decisions on different management options. However, it does not reveal the economically optimal path (e.g., the number of animals to remove per year at minimal management cost) to carry out the management plan. Further research could therefore be conducted to find these optimal paths using dynamic programming techniques (*Burnett, Kaiser & Roumasset, 2007*; *Hauser et al., 2007*; *Leung et al., 2002*). Decision variables in the context of IAS are most strongly influenced by the geographic area over which management is undertaken (*Robertson et al., 2015b*). These dynamic programming models allow the optimisation of an objective function (e.g., the sum of discounted management and damage costs) under various constraints but are harder to solve mathematically (*Hauser et al., 2007*). These models are more complex when aiming at comparing different management options or combining different management options in a single model because costs and benefits differ by management option. They allow however to economically optimize the management approach.

Our results have broader implications for conducting CBA for IAS management approaches. First, performing CBA requires identification of species impacts and the quantification of those using standardized information available. However, although we were able to place a negative value on an individual bird applying costs for eutrophication and agricultural damage, other costs may not be scalable in the same way (e.g., conservation impacts). Second, management costs do not always relate to the number of individuals of a species. For example, we showed the management costs for moult capturing geese

only varied when a certain threshold number of geese are caught. Third, CBA can be very informative for management decisions, but is often complicated, requires impact types that can be quantified in monetary terms, straightforward population models and may require many assumptions. Cost-benefit analysis might also be more appropriate for management of established species than for newly introduced species with limited information on population dynamics, costs and benefits. Finally, CBA requires good registration and documentation of the cost of management performed in the field.

## CONCLUSIONS

The aim of this paper was to apply a bio-economic model in a cost-benefit framework to an IAS. We used Canada goose, *Branta canadensis* L. as a model species as this species is known to exert severe pressures on the environment and the economy in the study region. We compared a business as usual scenario with a management scenario where these were supplemented with additional coordinated geese moult captures. Our analysis shows CBA to be a valuable framework in support of decisions on IAS management as it supplements risk assessments. It provides a technique to integrate both ecological and economic effects in the decision process on managing biological invasions. Our CBA showed that, under the assumptions of the model, the damage that can be prevented applying additional coordinated moult captures outweighs the extra costs involved. Therefore coordinated moult captures should be considered as an additional management tool whenever the management objective is to limit the negative economic impact of *Bc* at a regional scale. Although every CBA approach has its limitations and assumptions to be met, we believe the large discrepancy between the business as usual scenario and enhanced (BAU + coordinated moult capture) scenario indicates a robust conclusion. This study has shown that it is possible to carry out CBA despite limited data availability. However, we recommend using available national or regional guidelines on CBA to ensure comparability.

## ACKNOWLEDGEMENTS

We thank Karel Van Moer (RATO vzw) for providing cost estimates of moult capture projects, time series of goose captures and additional information. The Agency for Nature and Forest and Inagro vzw provided data on their goose captures. We thank Sander Devisscher for data handling. We are grateful to Thierry Onkelinx for providing information on the population estimates of Canada goose based on waterbird census data. We thank Dr. H Schekkerman (SOVON Vogelonderzoek Nederland) for information and references on agricultural damage by geese in the Netherlands.

### Funding

This work was supported by the Interreg IV-A projects Invexo (Invasive Species in Flanders and the southern part of the Netherlands, 2009–2012) and Interreg 2Seas RINSE (Reducing the Impact of Non-Native Species in Europe, 2012–2014), co-funded by the

European Regional Development Fund (EFRO). The funders had no role in study design, data collection and analysis, decision to publish, or preparation of the manuscript.

### Grant Disclosures

The following grant information was disclosed by the authors:
Interreg IV-A projects Invexo (2009–2012).
Interreg 2Seas RINSE (2012–2014).
European Regional Development Fund (EFRO).

### Competing Interests

The authors declare there are no competing interests.

### Author Contributions

- Nikolaas Reyns and Tim Adriaens conceived and designed the study, performed the experiments, analyzed the data, wrote the paper, prepared figures and/or tables, reviewed drafts of the paper.
- Jim Casaer, Koen Devos, Frank Huysentruyt, Peter A. Robertson and Tom Verbeke reviewed drafts of the paper.
- Lieven De Smet conceived and designed the study, analyzed the data, reviewed drafts of the paper.

### Data Availability

The data on population estimates and agricultural damage used in this study, full bio-economic model, and R code for analysis and graph production have been uploaded as Data S1.

### Supplemental Information

Supplemental information for this article can be found online at http://dx.doi.org/10.7717/peerj.4283#supplemental-information.

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
