# Peer review of "Cost-benefit analysis for invasive species control: the case of greater Canada goose Branta canadensis in Flanders (northern Belgium)"

_PeerJ, doi:10.7717/peerj.4283_

## Round 0.1 · original submission · Minor Revisions

Both reviewers have done an excellent and detailed review of your paper.

You have to pay special attention to the structure of the text, clarify the use of some terms such is efficiency, effective, economically optimal, and cost effectiveness as synonyms while they are not. Also, is pretty important to define the geographic scope.

In Methods, you have to be clear defining which costs and benefits are assessed. There are also some editorial corrections and suggestions.
I am pretty sure the paper will improve if you follow carefully these suggestions.

·

Basic reporting

Following the guidance of PeerJ I have checked figs and tables, the raw data and language. The raw data and code can be opened, and are well described in English. Language is clear, intelligible and professional. At a few places in the text minor details were found that can easily be corrected in a final stage. The references to literature provide sufficient context to Invasive Alien Species management, environmental economics and the model species. The structure of the article is professional. There are no hypotheses or research questions, but there is a clear aim and relevant results in relation to that.

Experimental design

The study reports on original primary research within Aims and Scope of the journal. A research question or an hypothesis is not provided. However, in my view, there is no need for them given the aim of the paper to demonstrate comparison of scenario’s using a bio economic model. The application of such a model is relevant & meaningful and fills an identified gap. The study uses techniques that suit the purpose. There are no ethical issues that could be an objection to publish the study. The methods are described with sufficient detail & information to replicate the study, provided the reader has a good mathematical background. Below, I will identify how the authors may facilitate readers that are less proficient in mathematics.

Validity of the findings

I appreciate the study as a worthwhile exercise. The data that are used refer to goose census data, harvest data and costs of water sanitation and crop damage. The authors rightly point out that data availability is limited. Still, they are able to put it to good use with their model. The data have not been gathered for this particular purpose but they are used in a logical way and the analyses are statistically sound. The parameters that feed into the models can be regarded as good approximations and the conclusions will certainly be robust for changes in parameter values. The conclusions are well stated, linked to the original aim and limited to supporting results.

Additional comments

By applying this model, it becomes clear under what assumptions the enhanced scenario of population management can be economically justified in comparison to business as usual. This is a general advantage of using (quantitative) models and it also works well in this case. The principal point that application of a Cost Benefit Analysis provides a useful framework to support decision making is well-elaborated.

Maybe it would be good if the authors mention explicitly why an alternative scenario of ‘enhanced scaring of geese’ is not considered. The value of a Decision Support System is greater when all relevant alternatives are considered. In theory, goose damage may be prevented by scaring geese. Probably, the authors have reason to believe that scaring is not effective in practice, rendering the ‘enhanced scaring of geese’ scenario ‘not relevant’?

The sensitivity analysis is restricted to exploring variation in a few relevant parameters (observed capture rate, discount rate and unit costs for eutrophication). The authors have made a selection for parameters they consider relevant. It thus remains unclear whether variation in other parameters may affect the conclusions and what the order is in parameter-sensitivity. For example: The parameters of the logistic growth model have been derived from limited census data. Thus, the value for carrying capacity (K) will be biased if the census results for the last years in the time series are biased. It would be good if the authors could explain that the results are robust for variation in the value for r and K and the unit price of hay.

In words the authors have identified what the main assumptions are and to what extent they can be considered justified. One of these is the assumption that costs of maintaining a constant removal/capture rate are constant over a range of population densities. For this species, under these conditions, I think it is fair to make this assumption. Still I would suggest stating explicitly in the text whether the model conclusions will be sensitive for a change in model formulation with regard to these costs. Am I right to believe that this will not be the case, because costs of management are small relative to damage costs?

Personally I would have appreciated more insight in the (calculation of the) population development under the two scenario’s and the observed field data from 2010-2015. From the text (lines 447 and 328) I read that “the predictions from this model were close to the observed results of a subsequent eradication “ and “at a lower than observed capture rate Bc could not be eradicated within the time horizon “. Is it possible to plot the observed data after 2009 in fig 3 with different symbols? Please consider adding more information on the predicted population development under the two scenarios (either as a graph or in words).

From the information provided (line 276 “...the population level before moult capture in year t equals the remaining post-moult capture population in year t-1 projected one year ahead on the logistic growth curve”), the principle of the calculation of predicted population trajectories is clear. It does, however, require some mathematics to actually implement that into the formula for the growth curve provided (equation 1; in that formula the reference is the start population). The reader will be facilitated if the authors indicate that formula 1 can be used to calculate the population level before moult capture in year t, by replacing p0 by the remaining post-moult capture population in year t-1 and by replacing t by 1. Maybe it is also helpful to provide the script for the full bio-economic model.

The authors point out that the benefits of management may be offset by increase in abundance of other species of geese (native or non-native). Would it be good to provide a little more background information on the status and numbers of relevant species in this respect in the introduction? What proportion of current damage is caused by Canada geese and what by other species?

Given my proficiency in English, I am not in a position to comment upon language and grammar. But to me the text is entirely clear. Below, I present the scant editorial comments that I think are useful to make.

In line 174 remove the word ‘and’ (double word)
Line 244-246. It appears as if information is repeated? Please rephrase
Line 447-449 I do not entirely understand the meaning of the sentence “The predictions from…..” Could you please rephrase this sentence?
Table 1 line 3: a bracket points in the wrong direction

Reviewer 2 ·

Basic reporting

see below - general comments

Experimental design

See below - general comments

Validity of the findings

See below - general comments

Additional comments

Summary
This study presents a cost benefit analysis (CBA) of the management and control of an IAS in Flanders. The BAU scenario is compared with a scenario that includes coordinated moult captures in addition to the BAU actions. The results show that the enhanced scenario had higher benefits (estimated as avoided costs from damage to grasslands and avoided damage from eutrophication) than additional costs. Some sensitivity analyses were performed with respect to estimate of N-damage, capture rate and discount rate.

General comments
The question addressed in this study is an important one. Given a limited availability of funds and resources, biodiversity management options should be carefully selected and motivated. Cost benefit analysis provides a useful tool for making such motivated policy choices. While the case study is an interesting one, the current version of the text raises several issues.

Firstly, the structure of the text could be greatly improved. Most of the motivation regarding the assumptions that were made when performing the CBA were only mentioned in the discussion section, while they should be mentioned in the method section. Examples of these include the selection of damage types, the selection of parameters for the sensitivity analysis and the option not to include rebound effects. In addition, it is only in the section ‘sensitivity analysis’ that the study period is mentioned, while this should again be included in the method section.

Secondly, the text (especially the introduction) seems to use the concepts ‘efficiency’, ‘effective’ ‘economically optimal’ and ‘cost-effectiveness’ as synonyms while they are clearly not.

Thirdly, the CBA is not correctly described and implemented.
- In the introduction (lines 116-135) the authors mention that a CBA is used to search for the scenario with the ‘lowest total costs’, to select the scenario that ‘maximizes avoided costs’ or the scenario with ‘the highest net PV (=total discounted avoided loss minus total discounted management costs)’. These three statements are contradictory and only one is correct (the last one).
- In addition, when implementing a CBA it is important to clearly state the standpoint from which costs and benefits are assessed. The current study seems to perform a social CBA from the viewpoint of society in order to minimize the total net social costs associated with Bc management in Flanders. However, the text is not always consistent with this viewpoint: e.g. line 158 mentions ‘… do not represent a direct cost for the regional government or local authorities’ which seems to imply that the CBA is excluding costs from private citizens.
- In a CBA costs and benefits are valued based on opportunity costs and not based on market prices, since the latter can be distorted by taxes, subsidies, imperfect information and imperfect competition. Thus market prices are unlikely to reflect the true social value of resource. Yet in the current study, the damage to grasslands is based on market prices for hay (line 222) without correction for sales taxes and agricultural subsidies. In addition, the damage from N and P is estimated based on avoided costs (line 193-195) which is a convenient method but has serious disadvantages since it is typically leading to an underestimation.
- The geographic scope of the CBA should be mentioned explicitly and evaluated. Why look at Flanders and not the whole of Belgium? Are these geese population so local that they do not cross borders?
- I cannot support the conclusion that ‘limiting the scope or complexity of a CBA is advisable without rendering the analysis unrealistic (lines 519-520)’. The essence of a CBA is to include all costs and benefits to all affected parties. Even though not all these impacts can be valued in monetary values, they should still at least be mentioned and not dropped for convenience reasons. Moreover, the current study is not exactly a good example in this sense since both the costs and the benefits of management actions are underestimated (as the authors acknowledge themselves in the discussion section). When both costs and benefits are underestimated it is not clear how the total result (PV values) are related to the true total impact.

Specific comments
- Introduction. Line 133. Several other interesting applications of CEA or CBA to invasive alien species management could be mentioned and used in this study (see, e.g., Waigner et al. 2010 or Schou & Jensen, 2017)
- Line 141: please define ‘Anseriformes’ when you use the term for the first time.
- Line 172: How do you know these are ‘the main economic impacts’ if the other impacts are not valued?
- Line 193: The estimated range for the P-value is very high (80-800 euro/kg). Why is this range so large? Why did you decide to use only the lower estimate in the CBA?
- Line 211: How are these compensation payments determined? What is their typical level?
- Line 237-248: It is not clear how this ‘number of captures needed to reduce the Bc population’ is calculated. Especially the second step is confusing.
- Line 275: How realistic is the assumption that the gees population does not migrate?
- Line 411: ‘were’ instead of ‘ware’
- Line 429: by applying
- Line 514: by applying
- The datasets are not clear to use as an outsider:
o What is included in the agricultural data? How are these data determined? Sources? How are they used?
o Why are the Canada goose data not in chronological order?

References
Wainger, L. A., King, D. M., Mack, R. N., Price, E. W., & Maslin, T. (2010). Can the concept of ecosystem services be practically applied to improve natural resource management decisions?. Ecological Economics, 69(5), 978-987.
Schou, J. S., & Jensen, F. (2017). Management of invasive species: Should we prevent introduction or mitigate damages? (No. 2017/06). University of Copenhagen, Department of Food and Resource Economics.

---

## Round 0.2 · accepted · Accept

The authors have included all changes suggested by the reviewers and the editor